# A Potential Indicator for Assessing Patient Blood Management Standard Implementation

**DOI:** 10.3390/healthcare11162233

**Published:** 2023-08-08

**Authors:** Andrea Kazamer, Radu Ilinca, Stefan Vesa, Laszlo Lorenzovici, Iulia-Ioana Stanescu-Spinu, Ionela Ganea, Maria Greabu, Daniela Miricescu, Andras Biczo, Daniela Ionescu

**Affiliations:** 1CREST Association, 48 Alexandru Odobescu Street, 440069 Satu Mare, Romania; 2Department of Anaesthesia and Intensive Care I, Faculty of Medicine, Iuliu Hatieganu University of Medicine and Pharmacy, 8 Victor Babes Street, 400012 Cluj-Napoca, Romania; stefan.vesa@umfcluj.ro (S.V.); daniela_ionescu@umfcluj.ro (D.I.); 3Discipline of Medical Informatics and Biostatistics, Faculty of Dentistry, Carol Davila University of Medicine and Pharmacy, 4-6 Eforie Street, 050037 Bucharest, Romania; 4Faculty of Technical and Human Sciences, Sapientia Hungarian University of Transylvania, 4 Matei Corvin Street, 400112 Cluj-Napoca, Romania; lorenzovici@hospitalconsulting.ro; 5Discipline of Biochemistry, Faculty of Dentistry, Carol Davila University of Medicine and Pharmacy, 8 Eroii Sanitari Street, 050474 Bucharest, Romania; iulia.stanescu@umfcd.ro (I.-I.S.-S.); maria.greabu@umfcd.ro (M.G.); daniela.miricescu@umfcd.ro (D.M.); 6Department of Modern Languages, Faculty of Medicine, Carol Davila University of Medicine and Pharmacy, 8 Eroii Sanitari Street, 050474 Bucharest, Romania; ionela.ganea@umfcd.ro; 7Department Hamm 2 Manufacturing and Production Technology, Hamm-Lippstadt University of Applied Sciences, Allee 76-78, D-59063 Hamm, Germany; andras.biczo@hshl.de; 8Outcome Research Consortium, Cleveland, OH 44195, USA

**Keywords:** patient safety, indicator, patient blood management, Safety Index in PBM

## Abstract

(1) Background: Patient blood management (PBM) program as a multidisciplinary practice and a standard of care for the anemic surgical patient has an increasingly important role in reducing transfusions and optimizing both clinical outcomes and costs. Documented success of PBM implementation is not sufficient for implementation of recommendations and correct use at hospital level. The primary objective of our study was to define a composite patient blood management process safety index—Safety Index in PBM (SIPBM)—that measures the impact of screening and treating anemic patients on the efficiency and effectiveness of the patient care process undergoing elective surgery. (2) Methods: We conducted a retrospective comparative study in a tertiary hospital by collecting data and analyzing the Safety Index in PBM (SIPBM) in patients undergoing major elective surgical procedures. (3) Results: The percentage of patients from the total of 354 patients (178 in 2019 and 176 in 2022) included in the study who benefited from preoperative iron treatment increased in 2022 compared to 2019 from 27.40% to 36.71%. The median value of the SIPBM was 1.00 in both periods analyzed, although there is a significant difference between the two periods (*p* < 0.005), in favor of 2022. (4) Conclusions: Measuring the effectiveness of PBM implementation and providing ongoing feedback through the Safety Index in PBM (SIPBM) increases the degree to which opportunities to improve the PBM process are identified. The study represents a first step for future actions and baselines to develop tools to measure the safety and impact of the patient blood management process in the surgical field.

## 1. Introduction

Quality and safety standards have been developed to reduce medical errors and improve clinical outcomes [1]. Safety standards are developed to generate, through their continuous application, a reduction in medical errors and an improvement in healthcare [1]. Additionally, the Institute of Medicine demonstrates in a paper that performance evaluation and monitoring through internal or in-house tools improve the quality of care [2].

Some of the procedures with the greatest clinical risks are those requiring anesthesia and surgery. Complex preoperative planning is required to lower their potential risks, and therefore surgical and anesthetic safety standards (surgical checklists, patient blood management, and anesthetic safety standards) have been developed and have become internationally recommended practices [3].

The gap between identifying evidence-based innovations and recommendations and their regular and systematic application by healthcare professionals has been extensively researched [4]. As a result, research that strives to increase the pace of implementation of standards and best practices in real-world healthcare settings is given priority. Implementing any standard or organizational change presents major and broad problems. According to a worldwide poll of over 3000 executives, two-thirds of respondents said their organizations failed to achieve real performance improvement after implementing organizational reform [5]. Top priorities for implementation science and continuous development are identifying, creating, and measuring the impact of implemented solutions.

Anemia is a serious issue that affects 30 to 40% of patients prior to major surgery and is linked to greater erythrocyte transfusions, longer hospital stays, more frequent hospitalizations to intensive care, infections, thromboembolic events, and mortality. Surgical bleeding raises the need for transfusions, causes anemia, and independently raises mortality. Additionally, transfusion of allogeneic blood products is linked to greater morbidity and mortality as well as higher costs and limited resources. Hence, as a practical response, the idea of patient blood management was created and was published in the anesthetic literature as an editorial in Anesthesiology in 2008 [6].

The Patient Blood Management (PBM) standard specifically targets the risks of anemic patients undergoing major elective surgery. PBM is an evidence-based package of care to optimize patient medical and surgical outcomes through clinical management and preservation of the patient’s own blood [7]. Patient blood management is based on three pillars: diagnosis and treatment of anemia (especially iron deficiency anemia), reduction of blood loss, and avoidance of unnecessary transfusions [8]. This is in a setting with preoperative anemic rates ranging from 20 to 75 percent, with acquired anemia frequently being added to this ratio [9].

In addition to being a fundamental component of good clinical practice in transfusion, it is crucial for patient safety. With improved outcomes in terms of morbidity, mortality, quality of life, average duration of hospital stay, and patient safety, patients would stand to gain the most from patient blood management [10].

Transfusion should not be used as the first line of treatment for anemia for patient safety reasons. According to the research, the new standard of care should be a proactive, patient-centered approach to managing the patient’s own blood. The implementation of a multimodal PBM program (using the three pillars) results in a 39% decrease in transfusion rates, as well as statistically significant decreases in hospital length of stay and an 11% overall decrease in death rates. Globally, the overall number of problems is lowered by 20%. The number of acute renal failures is reduced by 26%, and the number of infections is reduced by 9% [11]. An unfavorable transfusion incident that could have been averted with adequate blood management of the patient could be regarded a professional error [12].

However, implementation of patient blood management is still far behind expectations of good and safe clinical practice, despite compelling evidence and ongoing policy momentum from the WHO (World Health Organization), practical guidance for healthcare providers and national authorities [13,14,15], and clinical guidelines and recommendations in many specialties and national health systems [16,17,18,19,20,21,22]. The PBM standard suggestions are more likely to face cultural and behavioral obstacles than actual evidence of their significance [23].

The studies regarding overall adherence to regulated standards have shown that it varies depending on hospital specifics, teams, patient safety culture, availability of resources at the health system or facility level, and staffing levels. On the other hand, according to studies on the application of quality and safety standards, such as accreditation standards, when there is regular measurement, monitoring, and verification at the clinical process level, adherence to the standard’s rules/measures rises, which results in better clinical outcomes [24].

Among the elements that accelerated or supported the implementation of the PBM standard, the collection of evidence and data is recognized to have a key role in the implementation of the PBM standard [23]. A related paradigm is using the audit and feedback strategy to assist doctors in identifying opportunities to enhance practice [25]. And, adjusting audits to the local environment is a requirement for audit and feedback success [26]. Among the measures that are proposed to be taken for the implementation of PBM is the measurement/quality assurance through the use of quality measures and the demonstration of PBM impact. Participants are motivated to learn knowledge and become a part of the change as a result of local research and quantification of results. Monitoring of indicators and a feedback system on the results achieved are tools that can support the implementation of standards and improvement of patient safety culture [10].

The aim of our study was to define a safety index of the patient blood management process—Safety Index in PBM (SIPBM). Using this monitored indicator, we can assess the level of adherence of a health facility to the PBM standard and measure the degree to which anemia remains undetected and therefore untreated in hospitalized patients. SIPBM measures the safety of the patient blood management process defined by correlating the degree of adherence to Pillar I of the PBM standard with its impact on the efficiency and effectiveness of the process [10,27]. Monitoring and critical evaluation of cost-effectiveness are vital for motivating employees and supporting PBM distribution and institutional implementation. If many studies confirm the reasons for the effectiveness of applying the PBM standard, cost analysis becomes challenging [11]. The relationship between the metrics used and their impact on the elements that drive costs would provide information that would motivate medical and management personnel to adhere to the PBM standard’s implementation.

The study represents a first step for future actions and a point of reference to develop tools to assess the efficiency and safety of the patient blood management process in the surgery.

## 2. Materials and Methods

### 2.1. Study Design

A retrospective comparative cohort study was conducted involving inpatients and elective surgery patients at the Institute of Gastroenterology and Hepatology Prof. Dr. Octavian Fodor, Cluj, Romania. The institutional database was consulted, and we included in our study patients undergoing major elective surgical procedures such as gastrectomy, duodenopancreatectomy and hepatectomy between 1 January 2019–30 June 2019, and 1 January 2022–30 June 2022. A total of 178 cases were identified in 2019 and 176 in 2022.

The time periods were selected in this manner because the PBM standard in Romania went into effect on 28 September 2018. Due to COVID-19, it was only in 2021 that interventions were undertaken in the institution’s anesthesiology and surgical departments to promote the PBM standard by conducting educational meetings on virtual platforms on the standard’s implementation. Several instructional seminars about the standard’s application at the institutional level were held on virtual platforms between the two studied periods, with surgeons and anesthetists from the hospital participating.

### 2.2. Inclusion and Exclusion Criteria

Patients were included in the study if they met the following criteria: (1) they underwent targeted surgical procedures: gastrectomy, duodenopancreatectomy, or hepatectomy; (2) they were hospitalized within the defined time intervals; (3) were scheduled for surgery. Patients who underwent emergency surgery were excluded from the study. The study was approved by the Institute’s Ethics Board (13960/22.12.2022).

### 2.3. Analyzed Parameters

Data for the 4 categories of study participant parameters were collected from the electronic patient data system.

The interventions carried out at the institutional level to stimulate the implementation of the PBM focused on the dissemination of information specific to Pillar I of PBM and on the development of measures, and internal procedures to ensure the implementation of the standard.

The Safety Index in PBM (SIPBM) is composed of specific data needed to identify anemic patients scheduled for surgery and the specific measures implemented to manage anemia and the risks generated by the presence of anemia, respectively, transfusion in anemic patients.

For the determination of the SIPBM, 4 categories of parameters and the scale for each category were defined as follows:

A. Independent parameters (IP) for detection of anemia (Table 1)

Hemoglobin level before surgeryTransferrin saturation index—TSATC-reactive protein (CRP)

B. Indicators of implemented measures (IM) (Table 2) to correct iron deficiency and treat anemia in patients at risk:Treatment with erythropoietin-stimulating agentsUse of transfusion protocol

C. Healthcare impact—HI (Table 3). The quantified outcomes are:Presence of transfusion-associated adverse events, allergic reactions: incidence of acute kidney injury (AKI) and infection rate, transfusion-related acute lung injury (TRALI), etc.Length of hospital stay (LOS)

D. Costs’ impact—CI—related to the care of patients undergoing major surgery. Given that the number of ICU days has the greatest impact on hospitalization costs, this parameter has been defined as number of ICU days (Table 4).

SIPBM was calculated as a composite index resulting from the parameters presented in Table 1, Table 2, Table 3 and Table 4, following the formula:$$\mathrm{SIPBM}=\frac{\mathrm{IP}}{\mathrm{IM}}\;\times \;\frac{\mathrm{HI}}{\mathrm{CI}}$$

### 2.4. Data Analysis

Statistical analysis was performed using the MedCalc^®^ Statistical Software version 20.218 (MedCalc Software Ltd., Ostend, Belgium). Quantitative data were tested for normality of distribution using the Shapiro–Wilk test and were expressed as mean and standard deviation or median and 25–75 percentiles. Categorical data were expressed as frequency and percentage. The chi-square test was used to test for differences in frequency and Mann–Whitney test was used for two group comparison of quantitative variables. A *p* value lower than 0.05 was considered statistically significant.

## 3. Results

In the defined study intervals, we identified through the electronic system a total of 2307 patients who underwent surgery in 2019 and 2254 in 2022, 403 going through the 3 specific mentioned surgical procedures. Out of these, a total of 354 eligible patients met the planned (elective) intervention criteria, classified into two groups: in 2019 (n = 178) and in 2022 (n = 176) (Table 5).

### Demographic Characteristics of the Patients

A comparison of the characteristics in the two study groups is shown in Table 6 and Table 7.

According to the results, there were no significant differences regarding the age structure of the two compared groups (2019 and 2022). If in terms of the age of the patients there were no major differences, in terms of the number of patients with more comorbidities there was an increase in 2022 compared to 2019 (Table 8). The difference regarding the percentage of patients with 10 or more comorbidities is significant, which increased by 14.34%, in 2022—26.70% compared to 2019 when it was 12.36%. (*p* < 0.001). In 2022, patients with 19 and 20 comorbidities were identified—2 cases each. The patients included in the study had the following types of anemia as comorbidities: iron deficiency anemia, iron deficiency anemia secondary to blood loss (chronic), anemia in neoplastic diseases, and refractory anemia. Diabetes mellitus, anemia in neoplastic illnesses, chronic ischemic heart disease, hypertension, nonspecific reactive hepatitis, jaundice, protein-energy malnutrition, etc., were some of the most prevalent comorbidities.

In order to refine the SIPBM indicator, the differentiated analysis and inclusion in the SIPBM construction of different types of anemia and the specific intervention method in their case was proposed.

Iron (Venofer-Vifor, France) was infused as 1000 (500 mg) mg dose in a slow crystalloid infusion (2–3 h) in 259 patients of the 2 groups (Table 9). Dose was adjusted depending on the iron level, severity of anemia, and pharmacy availability (shortage of iron products was registered depending on availability on the market from medical companies and financial resources of the hospital).

In 2019, intravenous treatment with iron was administered to 10 (23.26%) patients before the date of surgery out of the total of 43 patients who received treatment with iron products during hospital admission. Moreover, in 4 patients out of 10 (40%) who received iron, treatment was administered 7 days or more before surgery.

In 2022, treatment with Fe products was administered in 35 (67.31%) of the 52 total patients before surgery. In 5 patients out of those 35 (14.29%), treatment was administered 7 days or more before surgery.

The total number of male patients identified prior to surgery with hemoglobin levels less than 13 g/dL was 52 in 2019 and 49 in 2022 (Table 10), while for females (less than 12 g/dL) the total number was 21 in 2019 and 30 in 2022, which represents a significant difference (*p* < 0.01).

The SIPBM value was in the range of 0.083 to 4.

Although the median SIPBM value in the two compared groups, 2019 versus 2022, remained at 1000, the percentile value indicates a difference (Table 11). If in 2019 we can say that 25% of cases recorded SIPBM values of 0.5 or lower, in 2022 this threshold dropped to 0.333 of SIPBM. The variation also in the case of the 75th percentile shows an improvement in the SIPBM value (3000 in 2019 and 2000 in 2022), although in both periods it is above the stable process threshold, being above 0.5. The same difference in favor of the length of stay in the ICU in 2022 can be found if we conduct an analysis of the percentiles. Although the median value for ICU length of care is 2.8 in 2022 compared to 2.3 in 2019, the value for the 75th percentile is decreasing. A total of 75% of patients undergoing surgery spent a maximum of 4.975 days in the ICU in 2022, while in 2019 the maximum number of days of stay for 75% of patients was 5.675.

Moreover, according to the SIPBM analysis of the three categories of patients according to the type of intervention, there was a significant difference in the median value of the SIPBM indicator. This difference suggests a more stable and efficient process in patients who underwent hepatectomy interventions (Table 12). Only in the case of 25% of patients the SIPBM value registers values higher than 0.5. The value of SIPBM in the case of duodenopancreatectomy patients justifies the need for a clinical audit in order to identify the causes that determine the variations in the approach to patients, the efficiency and effectiveness of the process. More than 75% of duodenopancreatectomy patients recorded values above the safety threshold of 0.5 of the SIPBM indicator.

Post surgery complications as fistulas occurred in each procedure as follows: 6% in case of gastrectomies, 4% in case of duodenopancreatectomies, and 3% in case of major hepatectomies biliary.

## 4. Discussion

Delivering care that complies with the high standards demanded by contemporary health systems is becoming an increasingly crucial and demanding requirement for quality and safety. Therefore, health care professionals must plan and develop methods in place to align healthcare with the most recent scientific findings [28].

In the current study, we aimed to define a composite index to evaluate how closely a healthcare facility adheres to Pillar I of the Patient Blood Management standard and how this impacts the efficacy of care. Improvements in anemia management represent an opportunity to seek to improve patient outcomes as well as to reduce costs associated with hospital days and potential risks associated with blood transfusion.

Around 1.6 billion individuals [29] suffer from anemia, with the main contributing factor being iron deficiency. Patients undergoing surgery frequently experience anemia, which can be brought on by underlying diseases or surgical blood loss. Patients with preoperative anemia have worse surgical outcomes than non-anemic patients, according to numerous prospective and retrospective cohort studies. Early research revealed that postoperative mortality in patients undergoing different types of surgery was independently predicted by significant preoperative anemia (with hemoglobin levels < 8 g/dL). Moreover, patients who underwent major non-cardiac surgery and had preoperative anemia had a greater 30-day postoperative morbidity and mortality rate than those who did not have anemia [13,30].

Because it is thought to be safe and has a quick effect, transfusion may be considered the standard treatment for anemia. However, there is growing evidence that using transfusion to treat preoperative anemia carries hazards that could raise the risk of morbidity and mortality in surgical patients. As a result, new strategies for treating anemia are becoming more widely recognized and are known as patient blood management (PBM) [29,31].

When the Society for the Advancement of Blood Management was established, the phrase “blood management” evolved from being viewed as regulating the supply to blood banks to being viewed from the standpoint of the patient, thus as change in the perspective towards the patient taking place. PBM became the standard for goal-oriented patient care, being based on published evidence and best practice, with cooperative inclusion and patient empowerment whenever possible, with the purpose of enhancing clinical outcomes. The three pillars of focus for PBM in patients undergoing surgery are identifying and treating preoperative anemia, minimizing perioperative blood loss, and utilizing and optimizing the patient’s physiological reserve of anemia [32].

PBM represents the convergence and establishment of these approaches for using the most recent scientific research to conduct blood transfusion. The World Health Organization has accepted it in an effort to reduce unnecessary and inappropriate transfusions of blood and blood products while enhancing patient care. Specifically in surgery, the advantages of PBM seem obvious. In the surgical environment, significant direct and indirect cost reductions can be made, and preliminary data indicate that the application of PBM techniques is linked to better clinical results [32,33,34]. Nevertheless, PBM is not restricted to the perioperative situation, but is equally applicable to other therapeutic procedures during which significant blood loss is known to occur and transfusion of blood products is part of the recognized treatment [35].

On the other hand, the pharmaceutical treatment of perioperative anemia, which has a detrimental impact on patients’ health, is a key objective of a PBM-based strategy. In addition to an increased risk of postoperative problems (particularly infections), it is typically correlated with lengthier hospital stays and a lower survival rate. The results of patients are particularly impacted by postoperative anemia, which can result from a number of causes, including pre-existing anemia, perioperative blood loss, frequent blood collection, and insufficient nutritional intake following surgery. Iron supplementation is the main goal of a PBM-based strategy because iron insufficiency is a common postoperative symptom. In this situation, it has been discovered that administering intravenous iron, with or without erythropoiesis-stimulating drugs, is a secure and efficient method of treating anemia in a patient [36].

Nevertheless, it can be challenging to launch a PBM program, and assistance from the hospital’s management, IT teams, medical and nursing personnel, and transfusion service is necessary. To incorporate and coordinate important components in an efficient manner, a planned and organized strategy is required. The creation of a multidisciplinary team is one of the initial steps, and it is crucial to additionally include an IT representative [37]. It is crucial to use the data to teach doctors about stringent transfusion guidelines and clinical evidence from randomized controlled trials once the IT data are in place. After doctors have been taught, peer-to-peer comparison of compliance rates with the most recent recommendations is a useful strategy for promoting quality improvement and lowering unnecessary transfusions [37,38].

To assess the impact of interventions to identify anemic patients and correct anemia in hospitalized patients undergoing elective surgery on the effectiveness and efficiency of care, we used the Safety Index in PBM—SIPBM. In this retrospective comparative study, we discovered that actions taken to support the PBM standard within the institution, such as setting up educational meetings with surgeons and anesthesiologists via virtual platforms (Zoom), had an impact on the measures taken prior to surgery and increased the number of patients treated with iron. In the institution included in the study, the percentage of patients treated for anemia before surgery increased from 27.40% to 36.71% (a significant difference—*p* < 0.01) as a result of the implementation strategy suggested in the literature [4]. Only 40% of cases in 2019 and 14.29% in 2022, however, received iron administration at least seven days prior to surgery, as recommended. The fact that elective procedures are scheduled in a short period of time justifies these results [25]. Despite an increase in the proportion of patients whose anemia was treated prior to surgery, the median value of SIPBM remained at 1.00 in both research periods. This increase, combined with a decrease in the number of patients who received iron for at least seven days prior to surgical interventions, as recommended by PBM, and an increase in the number of days spent in ICU by most patients from 2.3 days in 2019 to 2.8 days in 2022, resulted in the SIPBM value remaining stable.

Although the minimal evaluation of anemia must include the complete blood count, the determination of serum ferritin, sideremia, serum transferrin, and transferrin saturation index (TSAT—indicator of iron usable for erythropoiesis) necessary for the evaluation of iron deficiency, and the determination of C-reactive protein (CRP—indicator of inflammation, necessary for the evaluation of anemia from chronic diseases) [39], within the studied group no data on the transferrin saturation index were available. Regarding the prevalence of anemia in surgical patients in the studied group 42.94%, this falls within the reported prevalence of anemia between 25 and 50% depending on comorbidities and geographical factors [40].

Baseline data, as well as SIPBM, are needed to measure the impact of PBM Pillar 1 implementation. In addition, data on trends in the PBM safety index will be important to track PBM Pillar 1 performance and monitor the impact of measures in hospital days, ICU days, and post-transfusion complications. Clinicians can test the correlation between the categories of parameters we propose, the number of anemic patients, and the measures they implement in this category of patients.

Depending on local conditions, data from a single center should be analyzed against all other national institutions of the same level of competence.

The PBM guideline implementation is hampered by a number of obstacles, and the recommended implementation solutions [4] call for integrating different approaches. The combination of information/awareness measures with those for keeping track of outcomes and giving feedback on progress is the most efficient. Whatever the strategy, the actions must become ingrained in the organizational culture and routine of the healthcare staff [16]. The day-to-day tasks of staff are impacted by the organization of a hospital-wide monitoring system that is user-friendly and offers real-time process information [10]. The SIPBM indicator can assist those implementing PBM standard measures by supporting the consideration of an implementation strategy adapted to specific institutional and local constraints.

An increased value for SIPBM implies a procedure with differences in the preoperative approach to the anemic patient as well as the development of some barriers. Identifying a barrier is insufficient to lead the selection of a method to promote standard adherence. To understand an issue thoroughly enough to enable the appropriate selection of options, the causes of each barrier must be identified and very explicitly related with the intended outcome to meet the performance objective [4].

The knowledge and beliefs about the intervention of the staff involved, the access to available knowledge and information, and the pressure for change may be factors that generate these variations in the process of approaching the anemic patient [25]. Recognizing the absence of a framework of local regulations, protocols, or policies that guide transfusion decision making as a barrier to access to knowledge and information. Specific protocols for a multidisciplinary approach to the preoperative anemic patient should be added to the institutional protocol for anemia therapy. The procedure should specify how procedure planning and patient assessments will be carried out in a multidisciplinary team. Other obstacles that hinder the adoption of PBM procedures include the institution’s structural characteristics and available resources [25]. Discontinuities in the provision of iron resources and products cause differences in the anemia management process. Thus, clinical audit testing of the effectiveness of PBM implementation trials and continuous feedback [26] by defining and monitoring the Safety Index in PBM (SIPBM) increases the identification of opportunities to improve the whole process of surgical patient care flow.

From the total of 23 items recommended by the Romanian PBM Guidelines [41] to be monitored by a hospital, we tried to define a comprehensive indicator for Pillar I that would support decision making on performance analysis and improvement of the PBM process. Simplifying the PBM performance monitoring process by reducing the number of indicators is argued by the fact that a large number of target indicators to track can paralyze the decision-making process [42].

However, defining how to calculate the Safety Index in PBM (SIPBM) requires further investigation. And, in particular, the introduction of new categories of parameters that define the patient’s condition (comorbidities) and have a direct impact on the level of efficiency of care. The Pillar II and III approach of the PBM standard combined with Pillar I measures provide additional insights into the patient’s blood management process.

The SIPBM indicator’s range of possible values is (0.2; 12). According to the outcomes thus far and the design of the indicator, as the SIPBM values increase, the process becomes less stable, with differences in practice, safety, and the effectiveness of the care process for anemia patients. Values greater than 0.5 may point to the need for a clinical audit to determine the factors that led to discrepancies in the treatment of patients. The reasons for this increase in the SIPBM indicator’s value include (i) the failure to treat anemia because anemic patients were not identified, (ii) the ineffective treatment of anemia through a partial adherence to the PBM standard, and (iii) potential complications during patient care due to anemia, transfusion, or other iatrogenic or unavoidable causes. The definition of an interval to define a safe and efficient care process will be ensured by the analysis of the causes of the variations and the values that the SIPBM takes in response to the variations of each parameter included in the composition of the SIPBM (IP—Independent parameters, IM—implemented measures, HI—Healthcare Impact, CI—Costs’ Impact).

Different metrics are used to assess the effect on PBM profitability. Time durations, the number of patients researched, patient groups, and PBM methods adopted are factors determining cost-effectiveness [11]. The days of hospitalization in ATI, which were the ones with the highest expenditures, were taken into consideration in our study for the computation of the SIPBM index. The quantification of the workload in hospital blood transfusion laboratories and the reduction of material costs and general human costs can be implemented for the improvement of the SIPBM indicator. The management of anemia through a cost-effectiveness lens is a research area and area of indicator creation.

Additionally, in order to overcome regional barriers and alter traditional attitudes toward transfusion monitoring, SIPBM provides easily observable data; yet, coordinated hospital-level actions are required to fully execute patient blood-flow management procedures. The use of SIPBM in conjunction with the analysis of transfusion opportunities [43] and the addition of additional parameters regarding hemoglobin concentration upon transfusion of blood components will increase the safety of the patient’s blood management process.

The SIPBM also provides additional information on current activities to identify and treat anemic patients prior to surgery. The evolution over time and comparison with previous values at the institutional level provides a more complete picture on the changes in the process of identifying patients with anemia and treating anemia preoperatively.

Our study has a number of limitations. The selected hospital cannot be fully representative for all hospitals in the Romanian health system. The institutional infrastructure and organization may have an influence on the evolution of the process and of SIPBM. Another limitation of the study is the comparative retrospective design that did not allow the creation of a true control group for testing the variation of the SIPBM. Additionally, another limitation is that the inclusion criteria in the study were the type of surgery and we are aware that this may give different results as compared with analyzing all types of surgeries. The extent to which our findings can be generalized or applied to other surgical or medical fields requires further investigation. Moreover, we did not assess postoperative complications between surgeries and in-between patients with or without iron corrections, as our retrospective study focused on the detection of anemia and increasing awareness about SIPBM and not necessarily on the impact/complications of anemia.

Extending the research over a longer period of time, patient populations, and assessing the degree of implementation of all the other three PBM pillars would provide more relevant information and the SIPBM index would become more meaningful. It is possible that differences in perioperative care, apart from PBM interventions, may have contributed to the obtained values of the studied SIPBM index. A meta-analysis of large, prospective, randomized controlled trials would be preferable to relate changes in parameters that may have a significant impact on the calculation and refinement of the SIPBM indicator.

## 5. Conclusions

Our study represents a first step for future research to develop tools to measure the safety of the surgical patients’ blood management process. The demonstrated advantages of PBM are not sufficient for its implementation in everyday life. Measuring the effectiveness of PBM implementation and providing continuous feedback through different indices like the Safety Index in PBM (SIPBM) increases the identification of opportunities to improve implementation in current practice of the PBM process. Further development of the SIPBM is needed for a full assessment of the impact on care costs and effectiveness.

## Figures and Tables

**Table 1 healthcare-11-02233-t001:** IP—quantification of independent parameters for detection of anemia on a scale from 1 to 4.

1. IP—independent parameters	Normal hemoglobin level > 13.0 g/dL (men), >12.0 g/dL (women)	1
	Hemoglobin level < 13.0 g/dL (men), <12.0 g/dL (women), TSAT < 20% and CRP > 10 mg/L	2
	Hemoglobin level < 13.0 g/dL (men), <12.0 g/dL (women), TSAT < 20% and CRP < 10 mg/L	3
	Hemoglobin level < 13.0 g/dL (men), <12.0 g/dL (women), TSAT unavailable and CRP < 10 mg/L	4

**Table 2 healthcare-11-02233-t002:** IM—quantification of indicators of implemented measures on a scale from 1 to 6.

2. IM—implemented measures	Hemoglobin level within normal limits	1
	Hemoglobin level below normal and received i.v.Fe at least 7 days prior to surgery—patient was not transfused	2
	Hemoglobin level below normal and received i.v.Fe no more than 6 days prior to surgery—patient was not transfused	3
	Hemoglobin level below normal, did not receive i.v.Fe and was transfused	4
	Hemoglobin level below normal, received i.v.Fe no more than 6 days before surgery and was transfused	5
	Hemoglobin level below normal, did not receive i.v.Fe and was not transfused	6

**Table 3 healthcare-11-02233-t003:** HI—quantifying healthcare impact on a scale of 1 to 3.

3. HI—healthcare impact	LOS ≤ the average length of hospitalization (for the period analysed), no complications (post-transfusion: infection, acute renal failure, death)	1
	LOS > the average length of hospitalization (for the period analysed) by maximum 2 days, without complications (post-transfusion: infection, acute renal failure, death)	2
	LOS > the average length of hospitalization (for the period analysed) by more than 2 days and/or complications (post-transfusion: infection, acute renal failure, death)	3

**Table 4 healthcare-11-02233-t004:** Quantification of cost indicators on a scale from 1 to 3.

4. CI—Costs’ impact	Value more than 30% higher than the median of hospitalization days in the ICU (for the analyzed period)	1
	Up to 30% higher than median of hospitalization days in the ICU (for the analyzed period)	2
	≤median of hospitalization days in the ICU (for the analyzed period)	3

**Table 5 healthcare-11-02233-t005:** Data regarding the surgical procedures included in the research.

			2019	2022	Total
Type of surgery	Duodenopancreatectomy	Count	45	67	112
		% within Lot	25.3%	38.1%	31.6%
	Gastrectomy	Count	79	77	156
		% within Lot	44.4%	43.8%	44.1%
	Hepatectomy	Count	54	32	86
		% within Lot	30.3%	18.2%	24.3%
Total		Count	178	176	354
		% within Lot	100 %	100 %	100 %

**Table 6 healthcare-11-02233-t006:** Demographic data (n = 354).

Characteristic		All Patients (n = 354) n (%)	2019 (n = 178) n (%)	2022 (n = 176) n (%)
Sex	F/M	132 (37.3%)/222 (62.7%)	59 (33.1%)/119 (66.9%)	73 (41.5%)/103 (58.5%)
Patients with hemoglobin level below 12.0 g/dL (women), respectively, below 13.0 g/dL (men)	Total	152 (42.94%)	73 (41.01%)	79 (44.89%)
	F/M	51 (14.41%)/101 (28.53%)	21 (11.80%)/52 (29.21%)	30 (17.05%)/49 (27.84%)
Transfused patients		19 (5.37%)	12 (6.74%)	7 (3.98%)
Total number of cases with acute renal failure		13 (3.7%)	5 (2.8%)	8 (4.5%)
Total number of deaths		18 (5,08%)	9 (5.06%)	9 (5.11%)

**Table 7 healthcare-11-02233-t007:** Data regarding the age of the patients included in the study.

	Sex	Number	Mean	Std. Deviation	Std. Error of the Mean
2019	F	59	62.44	12.970	1.689
	M	119	63.39	10.682	0.979
2022	F	73	63.52	10.975	1.285
	M	103	63.43	10.608	1.045

**Table 8 healthcare-11-02233-t008:** Comorbidities statistics in the patients included in the study.

	Number of Comorbidities									
	1	2	3	4	5	6	7	8	9	10 or more
2019	7	6	12	15	28	24	22	22	20	22
	3.93%	3.37%	6.74%	8.43%	15.73%	13.48%	12.36%	12.36%	11.24%	12.36%
	3.93%	7.30%	14.04%	22.47%	38.20%	51.69%	64.04%	76.40%	87.64%	100.00%
2022	1	6	11	16	17	23	20	20	15	47
	0.57%	3.41%	6.25%	9.09%	9.66%	13.07%	11.36%	11.36%	8.52%	26.70%
	0.57%	3.98%	10.23%	19.32%	28.98%	42.05%	53.41%	64.77%	73.30%	100.00%

**Table 9 healthcare-11-02233-t009:** Data regarding patients receiving intravenous iron (n = 354).

I.v. Iron	2019 (n = 178)	2022 (n = 176)	Total (n = 354)
NO	135 (75.8%)	124 (70.5%)	259 (73.2%)
YES	43 (24.2%)	52(29.5%)	95 (26.8%)

**Table 10 healthcare-11-02233-t010:** Situation of patients with hemoglobin levels below normal and iron i.v. administration.

		2019	2022	Total of Patients with Decreased Hemoglobin
Hemoglobin < 13.0 g/dL (men)		n = 52	n = 49	n = 101
	Iron was administered i.v.	15 (28.85%)	15 (30.61%)	30 (29.70%)
	Iron was not administered i.v.	37 (71.15%)	34 (69.39%)	71 (70.30%)
Hemoglobin < 12.0 g/dL (women)		n = 21	n = 30	n = 51
	Iron was administered i.v.	5 (23.81%)	14 (46.67%)	19 (37.25%)
	Iron was not administered i.v.	16 (76.19%)	16 (53.33%)	32 (62.75%)
Total patients with hemoglobin below baseline		n = 73	n = 79	n = 152
	Iron was administered i.v.	20 (27.40%)	29 (36.71%)	49 (32.24%)
	Iron was not administered i.v.	53 (72.60%)	50 (63.29%)	103 (67.76%)

**Table 11 healthcare-11-02233-t011:** Length of ICU admission and SIPBM.

	Year	Median (25; 75 Percentiles)
ICU days	2019	2.300 (0.900; 5.675)
	2022	2.800 (0.800; 4.975)
SIPBM	2019	1.000 (0.500; 3.000)
	2022	1.000 (0.333; 2.000)

**Table 12 healthcare-11-02233-t012:** Comparison of the SIPBM index between different types of surgeries.

	Surgery	25th Percentile	Median	75th Percentile
SIPBM	Duodenopancreatectomy	0.6667	1.0000	2.3000
	Gastrectomy	0.3333	0.5333	1.0000
	Hepatectomy	0.3333	0.5333	1.0000

## Data Availability

The data presented in this study are available on request from the corresponding author. The data are not publicly available due to privacy and ethical reasons.

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
