# Peer review of "A Potential Indicator for Assessing Patient Blood Management Standard Implementation"

_healthcare, 2023, doi:10.3390/healthcare11162233_

Round 1
Reviewer 1 Report
Dear Authors,
The paper is very pertinent and of great interest as it covers many patients.
From my academic background I would like to make a few comments.
I believe that sex should be used instead of gender since health indicators refer to sex and not gender. Gender belongs to the social and in this paper the social part of anemia is not studied.
I have not read exclusion criteria. I deduce that there are none. In case there are, please put them.
The construction of the index is well understood. But you should indicate the values that this index can take and how it is interpreted.
In the Data Analysis section, it is stated that mean, standard deviation, median and 25th-75th percentiles have been calculated.
The results are difficult to read. They should be clearer for the reader.
Author Response
Dear reviewer,
On behalf of our research group, we would like to thank you for your time and your comments. We highly appreciated your recommendations and taking into consideration your suggestions, we made the following changes to the manuscript:
- As the reviewer suggested, we replaced "gender" with "sex" in Table 6, line 223 of the revised manuscript.
- Regarding the exclusion criteria, we only had one, as mentioned on line 156. Patients who underwent emergency surgery were excluded from the study.
- The authors are grateful for the reviewer's recommendation to introduce more information about the values that the index can take and how it is interpreted, and a new paragraph was introduced on lines 433-445.
- We agree with the reviewer's observation about the difficulty in the data analysis section, and Table 11 (former table 8) (lines 265) was modified. Additionally, a paragraph was inserted below the table to provide better insight (lines 268-278). We hope that now the data is more accessible to the reader.
We are hopeful that the quality of the manuscript has been improved and that we fulfilled your requirements. Thank you very much for taking into consideration the publishing of our manuscript.

Reviewer 2 Report
This paper conducted a retrospective comparative study by collecting data and analyzing the Safety Index in PBM (SIPBM) in patients undergoing major elective surgical procedures. The manuscript is well-written. The introduction, rationale, methods, and results were presented in a clear and organized way. I only have minor suggestions for the authors.
1. Were all the included anemic patients diagnosed with iron deficiency anemia? Were there any other types of anemia? Please provide more details (ages, co-existing diseases, coagulation function, et al), if possible.
2. Please provide information of the percentages of patients receiving gastrectomy, duodenopancreatectomy or hepatectomy, and the blood loss during surgery, if possible.
Author Response
Dear reviewer,
On behalf of our research group, we would like to thank you for your time and your comments. We highly appreciated your recommendations and taking into consideration your suggestions, we made the following changes to the manuscript:
- As the reviewer kindly suggested, information regarding the age and co-existing diseases was introduced in the manuscript in Tables 7 and 8, lines 224-240.
- The authors are grateful for the reviewer's recommendation to introduce information of the percentages of patients receiving gastrectomy, duodenopancreatectomy or hepatectomy and a new table was inserted - Table 5 (line 218). Regarding blood loss, unfortunately, there were no data available in the database and thus could not be collected for analysis.
We are hopeful that the quality of the manuscript has been improved and that we fulfilled your requirements. Thank you very much for taking into consideration the publishing of our manuscript.

Reviewer 3 Report
The authors should be commended for their excellent efforts. However, a few points for their consideration are as follows:
1. Please mention the details of the intravenous iron administered - generic name, dose, duration of infusion, etc.
2. Could the authors analyze the differences between those who have undergone different surgeries? that is, between those who have undergone gastrectomy, duodenopancreatectomy and hepatectomy separately?
3. Why wasn't any clinical adverse reactions captured/included?
Author Response
Dear reviewer,
On behalf of our research group, we would like to thank you for your time and your comments. We highly appreciated your recommendations and taking into consideration your suggestions, we made the following changes to the manuscript:
- As the reviewer suggested, details of the intravenous iron administered were introduced in the manuscript on lines 244-248.
- The authors are grateful for the reviewer's recommendation to introduce date about the differences between those who have undergone different surgeries. At first, we focused mainly on preoperative detection of anemia and on awareness on this of both surgeons and anesthesiologists. Thus, we did not compare the outcome in-between different types of surgery or between patients with and without iron treatment. However, as the reviewer suggested, we introduce a new table (Table 12) on line 289 to compare the SIPBM index between different the types of surgeries analyzed in the study (lines 279-290).
- Regarding the reviewer's observation about the inclusion of the clinical adverse reaction, in the research methodology we only included adverse reactions related to the transfusion process were included in the research methodology, as our main focus was to track the impact of the application of the PBM standard. Previous studies have proven that the management of anemia in patients reduces the rate of transfusions and thus the occurrence of adverse reactions related to the transfusion process. This was why we watched for these types of side effects.
We are hopeful that the quality of the manuscript has been improved and that we fulfilled your requirements. Thank you very much for taking into consideration the publishing of our manuscript.

Round 2
Reviewer 3 Report
Thank you for the revision.
Minor edits are required.